# Query-based Image Captioning from Multi-context 360° Images

**Koki Maeda**[1,2] **Shuhei Kurita**[2] **Taiki Miyanishi**[3,2] **Naoaki Okazaki**[1]
[1]Tokyo Institute of Technology [2]RIKEN AIP [3]ATR
koki.maeda@nlp.c.titech.ac.jp, shuhei.kurita@riken.jp,
miyanishi@atr.jp, okazaki@c.titech.ac.jp

## Abstract

A 360-degree image captures the entire scene without the limitations of a camera's field of view, which makes it difficult to describe all the contexts in a single caption. We propose a novel task called Query-based Image Captioning (QuIC) for 360-degree images, where a query (words or short phrases) specifies the context to describe. This task is more challenging than the conventional image captioning task, which describes salient objects in images, as it requires fine-grained scene understanding to select the contents consistent with the user's intent based on the query. We construct a dataset for the new task that comprises 3,940 360-degree images and 18,459 pairs of queries and captions annotated manually. Experiments demonstrate that fine-tuning image captioning models further on our dataset can generate more diverse and controllable captions from multiple contexts of 360-degree images.

## 1 Introduction

Image captioning is a task of describing the context of an image in natural language (Vinyals et al., 2015; Xu et al., 2015; Lu et al., 2017). Existing image captioning datasets, such as Microsoft COCO Captions (Lin et al., 2014) and Flickr30K (Young et al., 2014; Plummer et al., 2015), are based on the conventional images with limited camera field-of-view. For such images, the contexts of the images are selective and even simplified because people tend to choose what to photograph and crop out. Such images often include a clear context, such as "*a bird flying over a body of water*" or "*a cat sitting on a mat*." Image captions in the existing dataset often follow such salient context, ignoring minor but non-negligible contexts of the image. This becomes particularly problematic in the case of 360-degree images captured by omnidirectional cameras, as they capture the entire visible context of the scenes indiscriminately and preserve rich minor details.

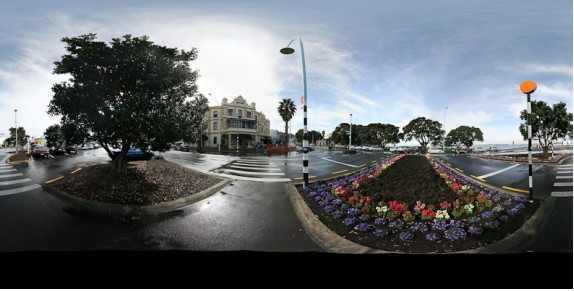

Figure 1: Overview of Query-based Image Captioning from 360° images, our new task is to generate relevant scene descriptions of images corresponding to linguistic information.

In this paper, we explore a novel **Qu**ery-based **I**mage **C**aptioning task for **360**° images (QuIC-360°). 360-degree images from omnidirectional cameras are indispensable in several fields, such as autonomous cars (Liao et al., 2022) and household robots (Srivastava et al., 2022). These images have no constraints on the camera field-of-view; therefore they capture rich contexts of the distant scenes. Indeed, multiple possible captions exist corresponding to the different aspects of the 360-degree images. Thus it is expected to generate textual descriptions that align with the user's interests and the relevant contexts when we develop systems that describe remote contexts from 360-degree images. Unfortunately, such rich contexts in 360-degree images are not fully expressed in the existing image captioning tasks. Conventional image captioning models concentrate on several salient contexts, ignoring other contexts that can be of in-

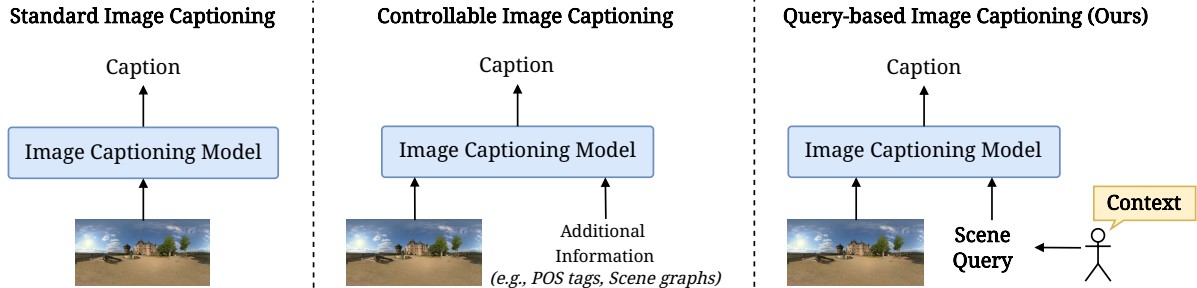

Figure 2: Comparison between image captioning tasks. Our task is one form of controllable image captioning for multi-context 360° images. In addition to the image, our task requires *query*, a word or short phrase to represent the intention of humans, as a signal.

terest to the users. To generate user-controllable image captions from multi-context images, we introduced textual *scene query* to the conventional image captioning task. Figure 1 presents an example of the 360-degree image, textual scene queries, and captions. The image encompasses various contexts, such as weather and surrounding traffic. It is impractical to mention all contexts in a single coherent human-readable caption. Therefore we introduce pre-defined scene queries that specify the contexts of the scene to describe. By querying the image, image captioning models can generate concrete captions from multi-context images.

To achieve the QuIC-360° task, we constructed a human-annotated dataset[1] for fine-tuning captioning models (Section 3). We collect 3,940 images taken by 360-degree cameras from Flickr and 18,459 manually annotated captions aligned with queries for our dataset. The captioning models leverage linguistic and visual information by training on our dataset involving different query-caption pairs for the same image.

We conduct experiments to assess our proposal dataset (Section 4). The quantitative results demonstrate that further fine-tuning an existing image captioning model with the proposed dataset improves the performance, controllability, and diversity in the QuIC task. Fine-tuned models also have the robustness to novel scene queries and out-of-domain images. In addition, the comparison of automatic evaluation metrics and human evaluation reveals that CLIP-based metrics are unsuitable for evaluating the QuIC task. We further investigate how models trained on the proposed dataset can generate captions aligned with given queries by qualitative analysis.

---

[1] https://github.com/Silviase/QuIC-360

## 2 Related Work

### 2.1 Image Captioning

Research on image captioning has made significant progress since the introduction of the Attention mechanism (Xu et al., 2015) for visual contexts. In particular, much attention has been drawn to controlling the described contents in captions in recent years. Controllable image captioning allows users to select and prioritize what should be described in the image (Cornia et al., 2019). Prior studies have explored methods to control the caption generation with information of different modalities in addition to the images, such as POS tags (Deshpande et al., 2018), sentiments (You et al., 2018), semantic roles (Chen et al., 2021), and abstract scene graphs (Chen et al., 2020a). Dense Captioning (Johnson et al., 2016) attaches detailed captions on each object in the conventional images. However, their approach often requires detailed and costly annotations and may overlook the coherent description of the entire scene. We explored visual context-disambiguation for multi-contexts images with QuIC by controlling caption generation with a short textual scene query that is easily interpretable by most users.

Several studies employ topic modeling typified by LDA and NMF, to automatically extract topics from the existing image-captioning datasets and utilize them in their proposed models (Yu et al., 2019; Mao et al., 2018; Al-Qatf et al., 2022). However, these studies rely on the existing image captioning datasets such as MS COCO Captions (Lin et al., 2014), which typically includes limited contexts in each image. We explore the scene captioning problem with images that have multiple contexts, and hence models require extrapolated scene queries to choose image contexts to be described. We prepare

the annotated captions following these pre-defined scene queries.

## 2.2 Language Conditioned Visual Task

In QuIC, models take an image and a short textual query as input. The input differences from existing image captioning methods are summarized in Figure 2. It is also considered that QuIC is similar to the existing visual question answering (VQA). However, as our task is based on image captioning, QuIC models expect more robust queries to scene images and generate more extended captions than those of VQA tasks.

**Visual Question Answering**. Visual question answering (VQA) is a task of answering a question given an image (Antol et al., 2015). It is similar to this research in that it focuses on local portions of an image and provides appropriate captions, and research on 360-degree images has also been conducted (Chou et al., 2020a; Yun et al., 2021). The answers of the VQA datasets are typically short, and VQA models often serve as a selection model from a pre-defined answer vocabulary set. Compared to the limited answer diversities in VQA, QuIC generates diverse scene captions following human-interpretable short queries.

**Referring Expression Comprehension**. Referring Expression Comprehension (REC) is another major branch of language-conditioned visual tasks. In REC, models specify the bounding box of the corresponding object following the textual *referring expression* as a "query" to the given image (Kazemzadeh et al., 2014; Plummer et al., 2015; Yu et al., 2016; Mao et al., 2016). This is similar to the visual context disambiguation of QuIC, although the output of REC is an object bounding box, not textual captions. Cirik et al. (2020) proposed the referring expression comprehension task on 360-degree images. Nowadays, several joint models of vision-and-language tasks, including VQA and REC, have been proposed (Chen et al., 2020b; Li et al., 2020). Among these models, we utilize the OFA model (Wang et al., 2022) for the joint visual and language input and textual output modeling baseline. Referring expression comprehension on videos is also an emerging task and has much attention because it is essential when we speciy some objects in video cameras (Li et al., 2017; Chen et al., 2019). Recently, Kurita et al. (2023) has proposed first-person vision-based referring expression comprehension of RefEgo, which

can be an important direction for real-world applications.

## 2.3 360°-image Dataset

For scene captioning, we search for general domain 360-degree image sets for our annotation. There are several existing 360-degree image sets. However, we notice that most of these datasets cover limited domains, such as indoor scenes (Cruz et al., 2021; Chou et al., 2020b; Chang et al., 2017; Zioulis et al., 2018; Karakottas et al., 2019; Zioulis et al., 2019), room layout sets and outdoor scenes (Sekkat et al., 2020; Liao et al., 2022), such as driving recordings. Datasets for saliency detection (Zhang et al., 2018; Hu et al., 2017) and gaze prediction (Lo et al., 2017; Fremerey et al., 2018; Xu et al., 2018) comprise 360-degree videos posted on video streaming service. Mazzola et al. (2021) collected 16 outdoor 360-degree videos for moving pedestrian recognition. We consider these datasets unsuitable for the general domain image captioning dataset because of the small number of videos. Some others are constructed by the integration or part of the existing image datasets (Wang et al., 2018; Cirik et al., 2020). We also refer to SUN360° (Xiao et al., 2012), a collection of 360-degree images from 360cities and the past de-facto standard 360-degree image dataset. However, SUN360° is not publicly available online on June, 2023.

## 3 QuIC-360° Dataset

We introduce the QuIC-360° dataset, which contains 3,940 panoramic images annotated with query-focused 18,459 captions. This is the first dataset that uses linguistic information to control the scene description for 360-degree images. We first describe the collection of 360-degree images from the Web and present how to annotate these images. We also present statistics of our QuIC-360° dataset.

### 3.1 Image Collection

We first collected 360-degree images of the diverse scenes for QuIC-360° and its annotation. Existing 360-degree image sets are often based on limited domains of scenes such as indoor (Antol et al., 2015; Cruz et al., 2021; Chou et al., 2020b) or outdoor (Liao et al., 2022). Therefore we newly assembled 360-degree images of various domains from Flickr, a popular community website for photo shar-

ing. We downloaded images from a group[2] sharing photos of 360-degree scenery views. On the Flickr website, multiple resolutions are available for some images. We chose the highest-resolution images for downloading. The total number of downloaded images is 12,930. However, these images often take different 360-degree image formats and include duplicated images for the same scenes. To improve the image dataset quality, we further filtered the downloaded images with the following steps: (i) To ensure consistency in image format, we utilized 360-degree images only in the equirectangular cylinder method. (ii) To filter similar images captured at the same location, we discarded images posted within an hour by the same user. (iii) To ensure image diversity, we sampled at most 100 images from the same user. (iv) To further improve the quality of the collected images, we manually remove inappropriate or duplicate images. After the filtering steps, we obtained 3,800 images in total.

Furthermore, we sampled additional 140 images by randomly selecting 20 images about each scene category (e.g., Restaurant and Shop) from Refer360° (Cirik et al., 2020), which are a part of the SUN360° (Xiao et al., 2012). We treated these images as a separate test set of the different image sources in the experiments. We annotated these images with captions and queries and called Refer-test.

## 3.2 Caption Annotation

In contrast to conventional image caption datasets, we annotated captions based on textual scene queries specifying a single context from the multi-context 360-degree images. In our task, queries play an important role in describing a part of the context from the 360-degree scene. When carefully selecting queries, we confirmed that the existing categories used for the image classification task were inappropriate as queries for the collected 360-degree image data. Therefore we pre-defined 34 scene queries suitable for describing 360-degree scenes by manually reviewing the collected images. See Table 2 for the pre-defined 34 scene queries.

We employed annotators using a crowd-sourcing service, Amazon Mechanical Turk (MTurk), to obtain captions following the pre-defined queries. For our task, each query for some image is expected to be relevant to that image and corresponding contexts. To this end, annotators chose three different

| Split | # Images | # Captions | # Vocab. | Avg. Length |
|---|---|---|---|---|
| Train | 3,000 | 9.438 | 8,090 | 20.2 |
| Valid | 400 | 1,251 | 3,098 | 20.2 |
| Test | 400 | 6,000 | 6,043 | 18.4 |
| Refer-test | 140 | 1,770 | 2,968 | 18.8 |
| Total | 3,940 | 18,459 | 10,862 | 19.4 |

Table 1: Dataset split and statistics in QuIC-360°.

| Query | # | Query | # |
|---|---|---|---|
| Location | 1,169 | Activity | 159 |
| People | 1,042 | Sea | 158 |
| Weather | 952 | Street | 146 |
| Building | 850 | Lake | 145 |
| Trees | 667 | Houses | 109 |
| Arthitecture | 523 | Garden | 84 |
| Landscape | 492 | Happenings | 84 |
| Interior | 398 | Corridor | 66 |
| What they are doing | 342 | Paintings | 53 |
| Cars | 296 | Stores | 53 |
| Small objects | 272 | Trains | 41 |
| Furniture | 243 | Fashion | 37 |
| Mountains | 223 | Animals | 35 |
| Plants | 197 | Traffic | 33 |
| Art | 195 | Foods | 24 |
| Rivers | 163 | Planes | 22 |
| Monuments | 161 | Accidents | 4 |
| | | Total | 9,438 |

Table 2: 34 scene queries and the number of each query selected in the train split of QuIC-360° dataset.

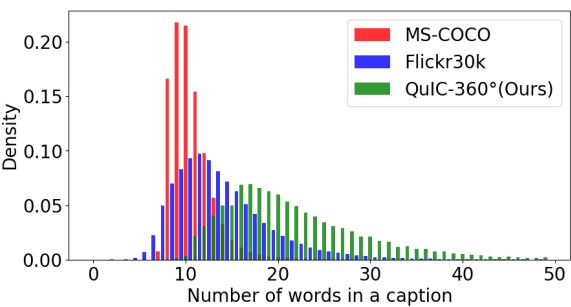

Figure 3: Distribution of caption length on MS-COCO, Flickr30k, and QuIC-360° datasets.

queries from the pre-defined 34 queries for each image and then wrote down captions related to each query[3]. To prevent generating brief captions, we asked MTurk workers to write captions no shorter than eight words. Then, we manually removed uninformative captions (e.g., "There is something on something in this image.") and obtained 18,459 captions.

---

[2]https://www.flickr.com/groups/360degrees/

[3]Appendix A presents details of the annotation website.

| Model | Fine-tuning | Query | # Vocab. | Len. | BLEU@4 | METEOR | ROUGE-L | CIDEr | SPICE |
|---|---|---|---|---|---|---|---|---|---|
| OFA$_{base}$ | ✗ | ✗ | 417 | 9.9 | 5.2 | 8.8 | 23.4 | 8.5 | 6.5 |
| | ✓ | ✗ | 395 | **13.2** | 10.2 | 11.3 | 28.0 | 14.3 | 7.2 |
| | ✓ | ✓ | **674** | 13.0 | **15.7** | **15.6** | **34.8** | **26.9** | **12.4** |
| OFA$_{large}$ | ✗ | ✗ | 521 | 10.9 | 5.9 | 9.4 | 24.7 | 10.1 | 6.9 |
| | ✓ | ✗ | 405 | 13.1 | 10.0 | 11.2 | 28.0 | 14.2 | 7.1 |
| | ✓ | ✓ | **590** | 12.0 | **16.2** | 15.3 | 34.7 | **25.0** | **11.8** |
| BLIP-2 (FlanT5$_{XL}$) | ✗ | ✗ | 455 | 10.7 | 8.1 | 10.2 | 26.1 | 12.5 | 8.1 |
| | ✓ | ✗ | **865** | 14.2 | 12.8 | 14.2 | 32.6 | 20.4 | 10.2 |
| | ✓ | ✓ | 796 | **18.4** | **13.1** | **15.8** | **33.8** | **23.0** | **12.3** |

Table 3: Comparison of query-based image captioning (QuIC) task performance using test split of the QuIC-360°dataset. All methods optimize the cross-entropy loss during finetuning.

## 3.3 Dataset Analysis

Table 1 presents the statistics of the QuIC-360° dataset. In total, our dataset contains 3,940 images and 18,459 captions for their related 34 queries. The vocabulary size of our dataset is 10,862, and the average caption length is 19.4, longer than existing image caption datasets such as MS-COCO and Flickr30k. We divided the QuIC dataset into train, valid, test, and Refer-test splits. The images in the test/Refer-test splits have at least four captions for each query to properly evaluate the caption generation performance.

Table 2 lists the frequency at which the annotators selected each query. The top six queries account for 55% of the total. We assume that these queries are easy to extract the context to describe the scene from the 360-degree images. We consider these six queries *major* and the rest *minor*. In the experiment, we observe the captions on minor queries generated by a model trained with only captions on major queries to validate the robustness to unknown queries (see Sec 4.3).

Figure 3 depicts the distribution of the caption length for MS-COCO (Lin et al., 2014), Flickr30k (Young et al., 2014), and our QuIC-360° dataset. The distribution of existing datasets peaks at around ten, while about half (48.5%) of the captions in QuIC-360° have over 20 words. This indicates that the captions of our dataset are longer than existing sets and have detailed information about images compared to existing datasets.

## 4 Experiment

We validate that the model learning with the QuIC-360° dataset surely enhances the controllability of scene captioning with baseline models. We use 3,000/400/400 images and 9,438/1,251/6,000 corresponding query-caption pairs in the train/valid/test

split.

## 4.1 Baseline Model for QuIC

To ascertain the effect of additional linguistic information in the form of queries alongside input images, we compared our method to the vanilla image captioning models of OFA (Wang et al., 2022) and BLIP-2 (Li et al., 2023) as baselines. Both baseline models exhibit high performance in conventional image captioning tasks. We gave these baseline models both an image and a modified prompt with scene queries as input to perform the query-based image captioning. However, we also noticed that these off-the-shelf models use fixed textual prompts in the image captioning training, independent of the various possible queries. The fixed prompt may cause under-fitting for the QuIC task, which requires the ability to generate captions for diverse queries. We therefore fine-tuned these off-the-shelf models with our modified prompts including a scene query. To examine the contribution of the query and fine-tuning, we compared the performance in three distinct scenarios, (i) w/o QuIC fine-tuning, (ii) fine-tuning with QuIC-360° in the conventional image captioning way, and (iii) fine-tuning with QuIC-360° by the modified prompts that include scene queries. We used the cross-entropy loss for the sentence prediction, following the conventional image captioning training of OFA and BLIP-2.

## 4.2 Metrics

We measured the similarity to human-written captions using BLEU (Papineni et al., 2002), METEOR (Denkowski and Lavie, 2014), ROUGE (Lin, 2004), CIDEr (Vedantam et al., 2014), and SPICE (Anderson et al., 2016). In addition, we used CLIPScore and RefCLIPScore (Hessel et al., 2021) metrics, which have been found to have the

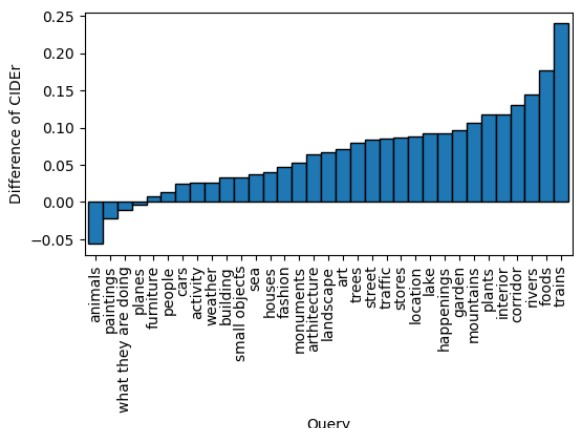

Figure 4: The difference of performance (CIDEr) in QuIC task per query between $OFA_{base}$ fine-tuned with/without queries.

| Method | Fine-tuning | Query | Relevance |
|---|---|---|---|
| Human | - | - | 3.87 |
| $OFA_{base}$ | ✗ | ✗ | 3.40 |
| | ✓ | ✗ | 3.40 |
| | ✓ | ✓ | **3.70** |
| BLIP-2 ($FlanT5_{XL}$) | ✗ | ✗ | 3.52 |
| | ✓ | ✗ | 3.64 |
| | ✓ | ✓ | **3.69** |

Table 4: Human evaluation of model outputs for image-query pairs sampled from the test split of the QuIC-360°.

| Model | Fine-tuning | Query | CLIP-S | RefCLIP-S |
|---|---|---|---|---|
| $OFA_{base}$ | ✗ | ✗ | **0.710** | **0.729** |
| | ✓ | ✗ | 0.661 | 0.707 |
| | ✓ | ✓ | 0.646 | 0.725 |
| $OFA_{large}$ | ✗ | ✗ | **0.733** | **0.730** |
| | ✓ | ✗ | 0.654 | 0.701 |
| | ✓ | ✓ | 0.648 | 0.729 |
| BLIP-2 ($FlanT5_{XL}$) | ✗ | ✗ | **0.734** | **0.747** |
| | ✓ | ✗ | 0.661 | 0.733 |
| | ✓ | ✓ | 0.671 | 0.739 |

Table 5: Comparison of the query-based image captioning (QuIC) task performance by CLIPScore and RefCLIPScore.

highest correlation with human assessments in conventional image captioning tasks. These metrics evaluate the similarity between the holistic context of an input image and a given caption using the CLIP model (Radford et al., 2021). Since the proposed QuIC task must focus on the local contexts of the image, we also confirmed whether these CLIP-based metrics are suitable for evaluating the QuIC task by using human agreement. Furthermore, we compared the average length and the vocabulary size of generated captions to examine whether a query enhances the diversity of captions (Wang and Chan, 2019).

## 4.3 Quantitative Analysis

We first evaluated the effect of introducing queries for 360-degree images. Then we conducted a human evaluation to test whether the models generate captions relevant to the queries. Finally we analyzed our model with out-of-domain settings and the effect of tuning specifically for image captioning (i.e., CIDEr optimization).

**Effect of Fine-tuning with Query.** Table 3 presents the results of query-based image captioning tasks on the test split of the QuIC-360° dataset. Models trained on our QuIC-360° dataset outperformed models that were pre-trained or fine-tuned in the conventional image captioning training on all metrics and successfully generated captions closer to human annotations. In addition, our fine-tuning method with queries significantly increased vocabulary size and the average length of captions compared to the pre-trained model. These results indicate that the introduction of queries helps cap-

tioning models to select the contexts of the images and improves performance on the QuIC task. We further validated whether the proposed models can generate diverse captions with qualitative analysis, detailed in Sec. 4.4.

**Performance Improvement per Query.** We examined the performance difference per query to determine which queries benefit from fine-tuning in the QuIC task. Figure 4 depicts the difference in CIDEr scores for each query on the test set between the model fine-tuned on the query-based image captioning and the conventional image captioning task. The result indicates a clear trend of improvements in most queries. We observe that queries with significant performance improvements provide captions not generated by standard image captioning. Although some queries showed performance drops, we assume this is due to the limited training data corresponding to these queries. This can be solved by collecting more diverse query-caption pairs on 360-degree images.

**Human Evaluation.** We conducted a human evaluation using MTurk to assess whether the outputs of models were relevant to given queries. First, we randomly selected 150 image-query pairs from the

| Training Data | # V | Len. | B@4 | M | R-L | C | S |
|---|---|---|---|---|---|---|---|
| ∅ | 399 | 9.8 | 5.7 | 9.0 | 24.4 | 10.2 | 6.6 |
| major | 537 | 13.0 | 14.8 | 14.6 | 33.7 | 25.6 | 11.1 |
| major + minor | 558 | 13.8 | 15.2 | 15.7 | 35.3 | 28.3 | 11.7 |

Table 6: Comparison of the query-based image captioning (QuIC) task performance on novel minor queries with different training set. The column colored in gray indicates the training set contains captions on the minor queries in the test set.

| Ft. | Q. | # V | Len. | B@4 | M | R-L | C | S |
|---|---|---|---|---|---|---|---|---|
| ✗ | ✗ | 206 | 10.4 | 4.3 | 8.5 | 22.7 | 7.8 | 6.8 |
| ✓ | ✗ | 236 | 14.0 | 7.5 | 11.2 | 27.0 | 12.9 | 8.1 |
| ✓ | ✓ | 418 | 14.0 | 14.1 | 15.2 | 32.9 | 22.9 | 12.5 |

Table 7: Comparison of the query-based image captioning (QuIC) task performance on the out-of-domain images using Refer-test split.

| Split | Opt. | # V. | Len. | B@4 | M | R | C | S |
|---|---|---|---|---|---|---|---|---|
| test | ✗ | 674 | 13.0 | **15.7** | 15.6 | 34.8 | 26.9 | 12.4 |
| test | ✓ | 749 | 17.7 | 15.0 | **17.5** | **35.3** | **30.5** | **13.3** |
| Refer-test | ✗ | 418 | 14.0 | **14.1** | 15.2 | 32.9 | 22.9 | 12.5 |
| Refer-test | ✓ | 457 | 17.9 | 12.6 | **16.5** | **33.1** | **26.2** | **13.5** |

Table 8: Comparison of the query-based image captioning (QuIC) task performance of OFA$_{base}$ with/without CIDEr optimization.

test split and generated captions corresponding to the image-query pairs by the image captioning models. Then, the workers rated the generated captions by a five-point Likert scale on how relevant these captions were to the given image-query pair. Table 4 presents the human evaluation of each model output with or without fine-tuning or a query. The best results for both OFA and BLIP-2 models are obtained when fine-tuned with a query, indicating that the proposed query-based image captioning can surely select the context aligned with a query from the image.

**Evaluation with CLIP-based scores**. Table 5 displays the query-based image captioning performance with the CLIPScore and RefCLIPScore. We found that both the models without fine-tuning or a query exhibited the highest values on the CLIP-based scores, while human and automatic evaluations exhibited the lowest performance as presented in Tables 3 and 4. This suggests that the CLIP-based scores are correlated well with humans in the standard image captioning task but not in the proposed task. Thus these metrics seem to be unsuitable for evaluating the QuIC task. We did not evaluate the performance with CLIP-based metrics in the subsequent experiments.

**Adaptability to Novel Queries**. Preparing queries that cover possible 360-degree images is a challenging requirement. We assessed the adaptability of our model to novel queries, which are not included in the training data. As introduced in Sec. 3.3, we sorted the queries by the number of captions and

treated the six largest queries as the major queries and the rest as the minor queries. In this experiment, we trained the OFA$_{base}$ model on the QuIC task with captions on major queries and evaluated it with those on minor queries.

Table 6 reports the result of the query-based image captioning performance on the minor queries in the QuIC-360° dataset with different training strategies. Although the model trained only with major queries performed slightly worse than the in-domain setting, it significantly outperformed the other model in all evaluation metrics without fine-tuning. The increased vocabulary and average length indicate that our model generates more diverse and controllable captions. These results imply that training on the QuIC task even with a limited number of queries enables the models to select the contexts from novel queries.

**Adaptability to Out-of-domain Images**. To evaluate our models' adaptation capacity to out-of-domain images, we utilized the Refer-test split, listed in Table 1. While test images of the QuIC-360° are collected from Flickr, images of the Refer-test (from Refer360°) are collected from 360cities. Table 7 compares the query-based caption generation performance on the Refer-test split. Despite the images being collected from different image sources, fine-tuning with the QuIC-360° on the QuIC task improved the performance in the Refer-test split compared to other strategies, with no fine-tuning and with fine-tuning on the standard image captioning task. In addition, the introduction of queries increased the vocabulary size and average length of generated captions. As with the in-domain results in Table 3, our query-based image captioning framework allows models to generate more diverse and accurate captions for out-of-domain images.

**Effect of CIDEr Optimization**. Empirically, optimizing the image captioning model using the test metric (i.e., CIDEr optimization) signif-

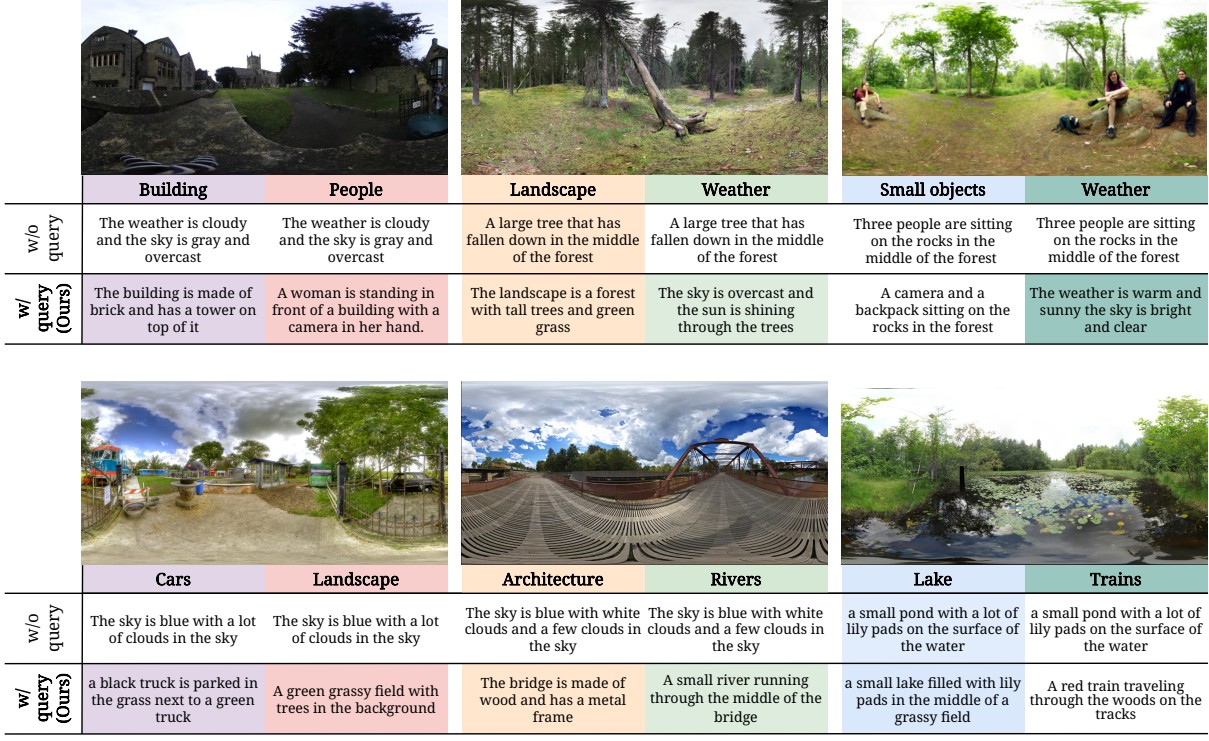

Figure 5: Qualitative examples. We colored examples that are relevant to the queries.

icantly improves the image captioning task's performance (Rennie et al., 2017). Table 8 presents the performance difference between models based on whether CIDEr optimization was conducted. Both models are trained on the QuIC task with our dataset. For our new task, the model tuned with CIDEr obtained higher values of automatic evaluation metrics for both test/Refer-test sets. It is also observed that models generated longer captions due to the CIDEr optimization. Thus the decrease in BLEU can be attributed to this. These findings confirm the usefulness of tuning models with test metrics for better performance in the QuIC task.

## 4.4 Qualitative Analysis

We compared the captions from models fine-tuned with queries to those from models fine-tuned without queries to clarify how the generated captions are diverse and consistent with the given query. Figure 5 illustrates examples of the query-based image caption generation[4]. The models trained without queries generated the same sentences regardless of the query provided, indicating that the vision-and-language model tuned with image-caption pairs only relied on image contents and described the

main content of the images. In contrast, the model tuned with image-caption-query triplets produced diverse captions based on the given query.

Although the models trained on the QuIC task can produce correct captions from 360-degree images in most cases, it sometimes generates incorrect captions. As depicted in the top-right and bottom-right examples, our model generated captions for non-exist objects related to a given scene query. For example, "camera" and "red train" do not actually exist in the top-right and bottom-right images, but our model wrongly gave descriptions including these objects following given queries. In future work, it is essential to address such hallucinations to ensure generating captions that correspond appropriately to the image-query pairs.

## 5 Conclusion

We have proposed a query-based image captioning for 360° images and constructed the QuIC-360° dataset for our new task, containing human-annotated captions focused on queries. In QuIC, models leverage textual queries to describe images with multiple visual contexts from 360-degree images. Experimental results show that further fine-tuning existing image captioning models with our dataset can improve the controllability and diver-

---

[4]We have attached other examples in Figure 9 in Appendix C.

sity of captions on 360-degree images. In addition, models trained on our dataset are robust to both unknown queries and out-of-domain images to some extent. Further investigation also reveals that CLIP-based metrics are unsuitable for the evaluation of our task, highlighting the target difference between QuIC and the conventional image captioning task.

## Limitations

**Images**. Our proposed dataset consists of images collected by web crawling. Although we gathered a diverse 360-degree images set including indoor and outdoor scenes, it still has a limitation in the coverage of the real world-scenes.

**Annotations**. We employed MTurk workers to obtain the captions for our QuIC-360° dataset. We provided clear instructions and examples for the workers and asked them to remove inappropriate or harmful captions. We also manually supervised to assure this. However, we do not ensure that all biases are completely removed.

**Others**. As mentioned in the qualitative analysis, the annotators tend to select queries from the photographed contents when they assign captions to images. Thus the models trained on our dataset might give wrong descriptions in the QuIC task when the image does not contain the object related to the given queries.

## Ethics Statement

We constructed this dataset based on the publicly available images posted on Flickr. We distribute the URL list to the Flickr images instead of the collected images themselves. Some of the images contain people, but we carefully removed captions that can be used to identify them. When used properly, our image and annotation dataset is beneficial as many other image datasets such as MS-COCO (Lin et al., 2014) and Flickr30K (Young et al., 2014; Plummer et al., 2015).

## Acknowledgements

This work was supported by JSPS KAKENHI Grant Number JP22K17983, and by JST PRESTO Grant Number JPMJPR20C2.

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

## Appendix

## A MTurk Annotation Details

We employed workers using Amazon Mechanical Turk to collect annotations for the 360-degree images. To ensure the quality, we hired those who resided in the U.S. and whose recent assignment approval rate was greater than 95%. Each annotation time was measured by a pilot task, and we paid rewards for the average time worker at the rate of $12 per hour. We have attached a snapshot of the annotation website in Figure 6.

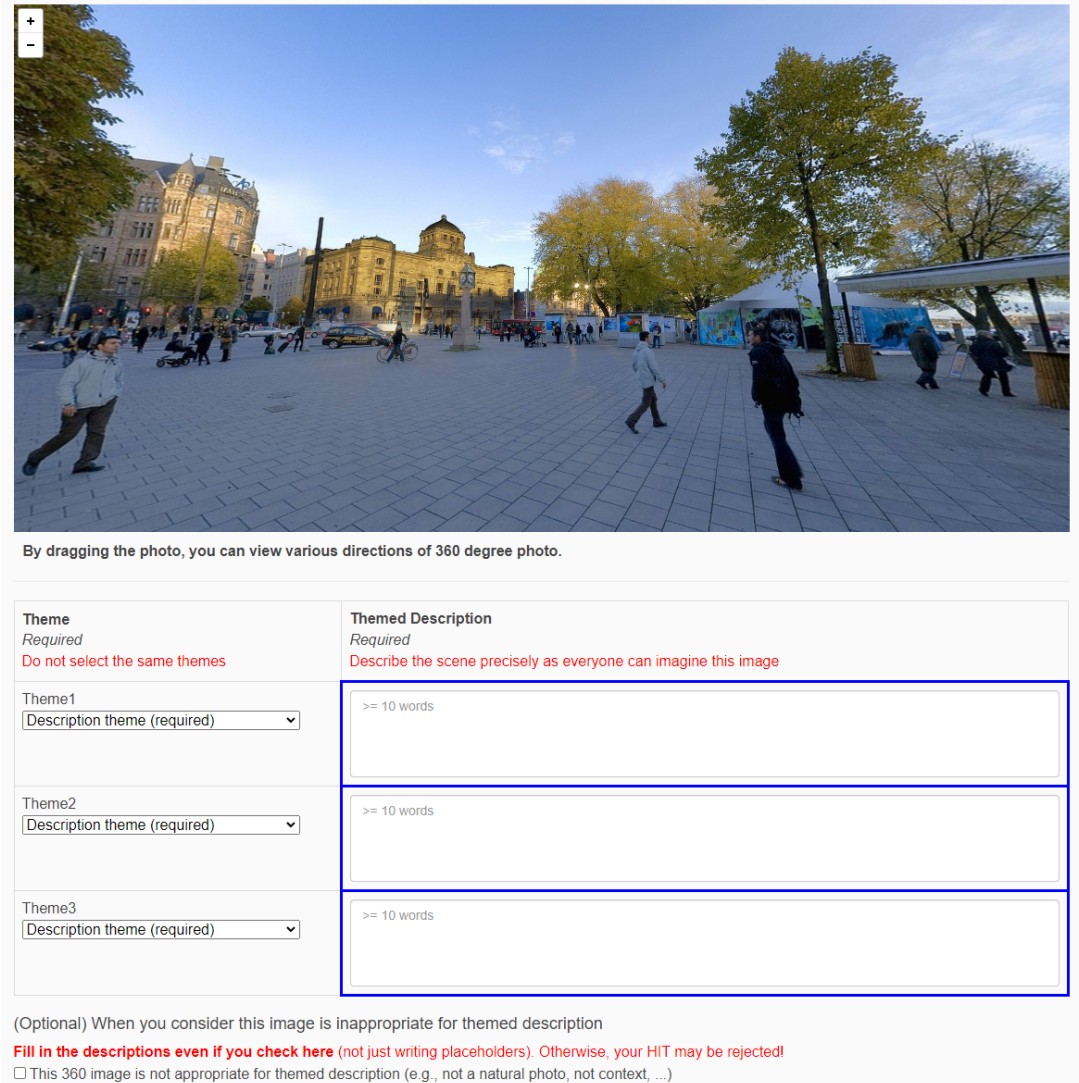

Figure 6: An screenshot of annotation in MTurk. The workers select three queries related to the presented images and give descriptions of each query.

## B Model Architecture

**OFA**. Figure 7 shows the illustration of our OFA architecture. OFA employs Transformer-based encoder-decoder architecture, which unifies various vision and language tasks such as image captioning, visual grounding, and visual question answering in a single architecture. In the image captioning task, OFA uses fixed prompts with images as textual input for the conventional image captioning task. Taking advantage of the prompt's characteristics, we modified OFA to be suitable for the QuIC task. We modified the original prompt to include the given scene query and fed it to the model to generate context-dependent captions.

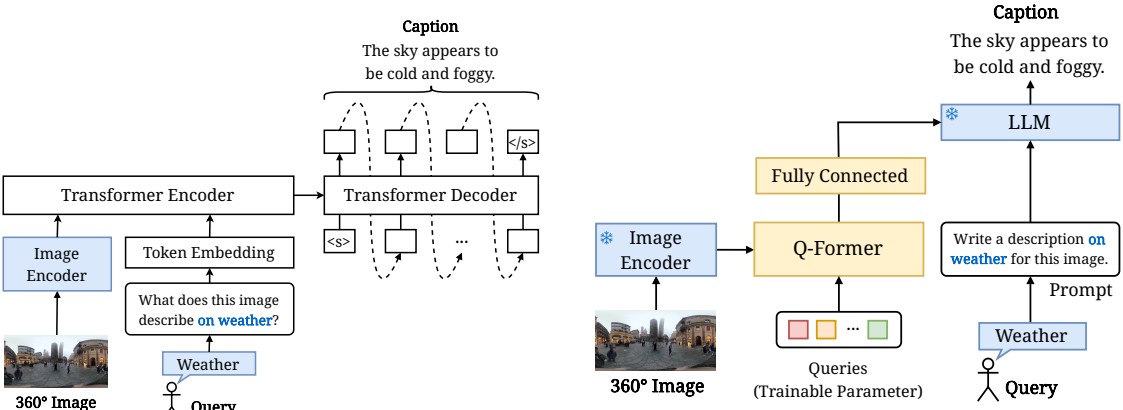

Figure 7: OFA architecture for the QuIC task.

Figure 8: BLIP-2 architecture for the QuIC task.

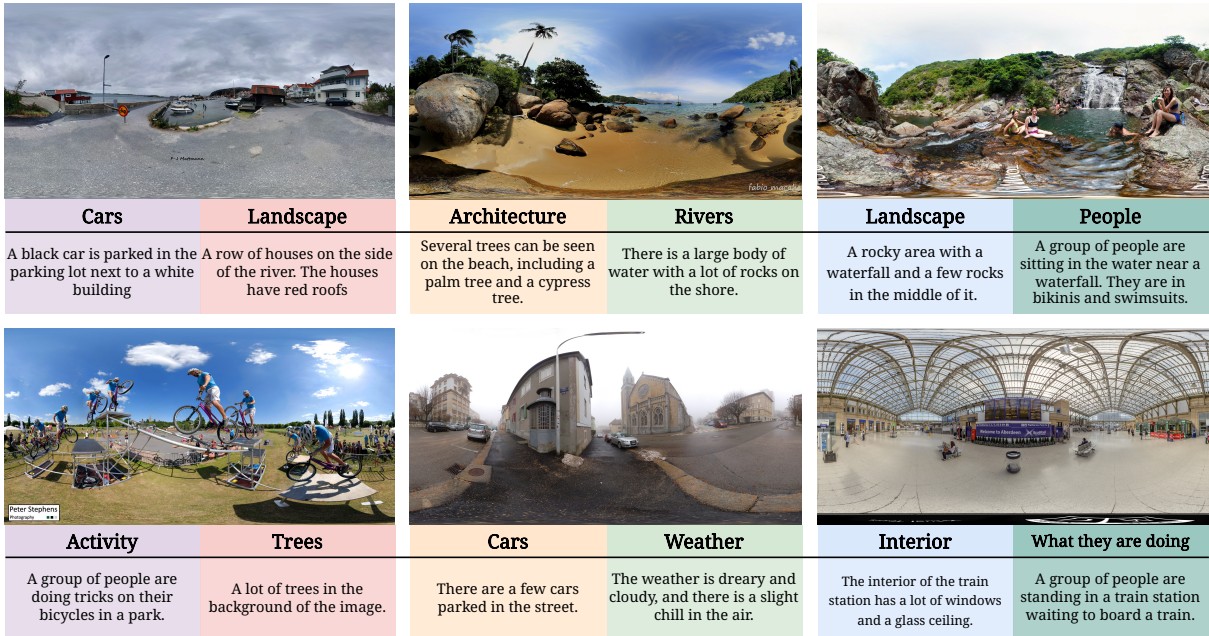

Figure 9: Additional qualitative examples generated by the models trained on our QuIC-360° dataset.

**BLIP-2**. Figure 8 delineates the architecture of BLIP-2. BLIP-2 integrates Querying Transformer (Q-former) to bridge the gap between frozen LLM and a pre-trained image encoder. Q-Former contains two transformer submodules for visual feature extraction and for processing textual input, which share the same self-attention layers. In the experiments, we used ViT-g/14 from EVA-CLIP (Fang et al., 2022) for the image encoder, and $FlanT5_{XL}$ (Chung et al., 2022) for encoder-decoder based LLMs. We added the original prompt containing the scene query to the model.

## C Additional Examples

Figure 9 illustrates additional examples produced by our method using the QuIC-360°dataset. We can observe that most outputs describe detailed information in response to the queries. The captioning model correctly recognized the materials of the ceiling and the fact that people are waiting in the station in the bottom right example. On the other hand, they fail to mention in detail when they do not acquire enough information from images. In the bottom left example, the description of trees gives less information about images. The model tends to generate such simple captions when the size of the object related to the query is small, as depicted in the example.

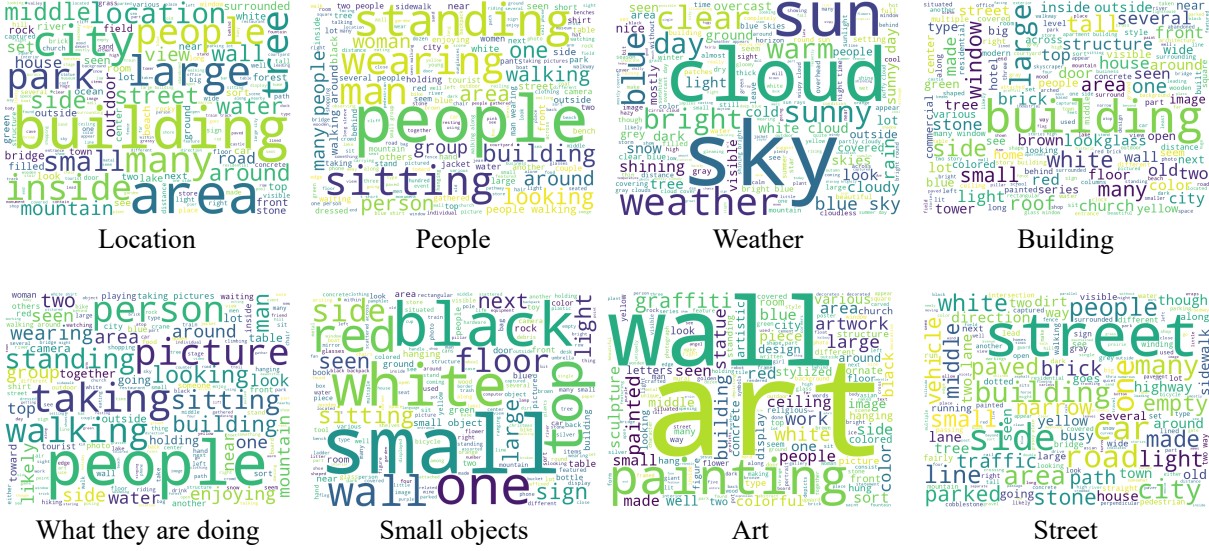

Figure 10: Visualization of word distributions of our QuIC-360° dataset by query.

# D Word Distribution

To clarify the context that the queries represent, we visualized word distributions for the four queries on each major and minor query in Figure 10. We can observe that each query contains the word itself and various words describing properties or actions. Interestingly, the caption for the query "Location" contains words related to other queries, such as "Building", "People", and "Trees". These queries are required to select contexts from multiple choices and to describe the most salient objects, which are more complicated.

# E Leveraging Non-language Data

We also tested a method to control the caption generation by once specifying the region to focus with a query and then using the obtained area as an aid. Specifically, we employed OFA for the reference expression comprehension task, and combined it with the Caption Anything Model (Wang et al., 2023)[5], which internally uses the Segment Anything Model (Kirillov et al., 2023). First, OFA model fine-tuned with RefCOCO Dataset predicts the bounding box from a given scene query and image. Then, pre-trained CA model generates captions based on the bounding box and images. We performed the same experiment on this pipeline, as presented in Table 3. However, CIDEr and CLIPScore were 4.5 and 0.670, respectively, the worst result despite using the state-of-the-art models.

---

[5]We used sam_vit_h for the segmenter, and Salesforce/blip2-opt-2.7b for the captioner as is.