# OpenReview forum: "Query-based Image Captioning from Multi-context 360° Images"
_EMNLP/2023/Conference — EMNLP 2023 Findings_

### Official Review · Reviewer_WuwK · 2023-07-28

**Soundness:** 4

**Excitement:**

3: Ambivalent: It has merits (e.g., it reports state-of-the-art results, the idea is nice), but there are key weaknesses (e.g., it describes incremental work), and it can significantly benefit from another round of revision. However, I won't object to accepting it if my co-reviewers champion it.

**Paper Topic And Main Contributions:**

The paper introduces a new task, Query-based Image Captioning for 360-degree images (QuIC-360). In this task, given a 360-degree panoramic image and a specific topic (a "query"), the model produces a caption that describes the aspects of the image related to the query. The paper also presents a novel dataset collected for this task and experimental results achieved by fine-tuning existing image captioning models on this dataset.

**Questions For The Authors:**

A. Many of the scene queries do not seem mutually exclusive: e.g. "cars" and "traffic", "activity" and "what they are doing", "interior" and "furniture", "art" and "paintings", etc. Would captions generated based on these very similar queries be sufficiently different? If so, in what way?

B. Why, do you think, are the CLIP-based metrics poorly suited for the data/task presented in this paper? Is it also the case for other types of controllable image captioning described in previous research, where only certain aspects of the image are described in the caption?

**Reasons To Accept:**

The paper presents a new take on controllable image captioning, and the dataset and reported results can become a benchmark for researchers interested in this topic. The performance of the fine-tuned models on the novel dataset demonstrate their potential for creating diverse captions focusing on different aspects of the images, given suitable training.

**Reasons To Reject:**

The task may not be sufficiently different from other types of controllable image captioning, guided by textual thematic "prompts", e.g. (Hu et al., 2022). The only obvious difference is the focus on 360-degree images but it doesn't seem from the paper that 360-degree images are (or should be) treated any differently from regular ones. The authors' claim is that 360-degree images contain more topics and details, but many regular images can also be quite detailed, and the examples provided in the paper (e.g. the forest, lake, bridge scenes in Fig. 5) are not much more detailed than non-panoramic images.

Hu, Y., Hua, H., Yang, Z., Shi, W., Smith, N. A., & Luo, J. (2022). Promptcap: Prompt-guided task-aware image captioning. arXiv preprint arXiv:2211.09699.


**Reproducibility:**

3: Could reproduce the results with some difficulty. The settings of parameters are underspecified or subjectively determined; the training/evaluation data are not widely available.

**Reviewer Confidence:**

3: Pretty sure, but there's a chance I missed something. Although I have a good feel for this area in general, I did not carefully check the paper's details, e.g., the math, experimental design, or novelty.

**Typos Grammar Style And Presentation Improvements:**

For clarity, it would be helpful to give an explicit definition of "(image) context" in Section 1 -- how this (generally vague) term is going to be used in this paper in particular.

31-32: "what to be photographed and cropped out" -> "what to photograph and crop out"

191: "comprises" -> "comprise"

260-261: "the existing categories used for the image classification task" -- which categories and task are referred to here? If it is a concrete previous work, it should be cited.

401-403: "We observe that queries with significant performance improvements provide captions not generated by standard image captioning" -- would be nice to illustrate this statement with examples, to make the claim more clear and convincing.

416-417: "how well these captions were relevant to..." -> "how relevant these captions were to..."

488: "whether" -> "based on whether"

---

> ### Author Rebuttal · Authors · 2023-08-29
>
> > Difference from other types of controllable image captioning, guided by textual thematic "prompts”.
>
> We have been aware of using “prompts” for guided image captioning in the course of our research. While both methods involve similar inputs and outputs, PromptCap requires much more detailed “questions” instead of our “queries” to control caption generation. This approach seems to require prior comprehension of the images, which deviates from our goal of automatic scene understanding.  We will cite this in our manuscript in the revision and clarify these differences.
> It is also notable that although the paper “PromptCAP”[2] was published in late 2022 to arXiv and was accepted in the ICCV2023, we do not necessarily assume it is worth mentioning as a part of “reasons to reject” in the EMNLP2023 review because we believe they were done in the same timeline.
>
> > Use of detailed regular images
>
> As a part of the scene captioning task where scene descriptions come from solely one 360-degree camera, we chose 360-degree images not only for their rich contexts but also because they capture entire scenes without selection bias. The contexts of normal images are almost always selective by photographers and even simplified because people tend to choose what to photograph and crop out. We stated this point in lines 24-41, and we will try to express more clearly the reason for choosing the 360-degree images in the camera-ready version.
>
> > Many of the scene queries do not seem mutually exclusive
>
> In this dataset, we selected topics that are frequently illustrated in scenery images. We do not impose that the topics are mutually exclusive and hence accept that some topics may overlap in the resulting generation. However, even in the topics that are apparently similar, we expect differences of the focused contexts, e.g., in the “cars” topic models describe the details of the cars while in “traffic” the models describe the road traffic states, including pedestrians and traffic lights. We allow these words overlapping in the query set to enhance the model’s robustness.
> We hope that the query set will become open-vocabulary and the captioning model will be able to generate captions that accurately capture the nuances of those related words, and our study serves as a dataset to facilitate this.
>
> > Reasons why the CLIP-based metrics poorly suited for our data/task
>
> A prior work [1] showed that CLIP-based metrics are sensitive to whether captions depict salient objects in the image. On the other hand, our proposed task requires captions to focus on various objects or scenes corresponding to the given queries.  In particular, totally different captions can be correct for the same image, depending on the queries. For this reason, CLIPScore, which treats holistic images and texts in the joint space, is poorly suited for our data/task.
>
> [1] An Examination of the Robustness of Reference-Free Image Captioning Evaluation Metrics, (Ahmadi and Agrawal, 2023)
>
> [2] PromptCap: Prompt-Guided Task-Aware Image Captioning (Hu et al., 2022)

---

### Official Review · Reviewer_4vda · 2023-08-04

**Paper Topic And Main Contributions:** 1. A new Image Captioning task and da…
**Soundness:** 3

**Excitement:**

3: Ambivalent: It has merits (e.g., it reports state-of-the-art results, the idea is nice), but there are key weaknesses (e.g., it describes incremental work), and it can significantly benefit from another round of revision. However, I won't object to accepting it if my co-reviewers champion it.

**Questions For The Authors:**

Can this dataset(QuIC), like SA[0], use "points" or bounding boxes as cues to generate corresponding captions?

[0] Kirillov A, Mintun E, Ravi N, et al. Segment anything[J]. arXiv preprint arXiv:2304.02643, 2023.

**Reasons To Accept:**

1.The QuIC dataset has been proposed, which to some extent, enhances the Image Captioning scene in the field of panoramic vision.

2.The paper conducted comprehensive testing and evaluation of the dataset from various perspectives, and presented the results of human evaluations.

**Reasons To Reject:**

1.The experimentation involved a limited number of models. For a new dataset, it would be beneficial to conduct evaluations on a more diverse set of architectures.

2.The dataset lacks certain mismatched cases, such as when the object corresponding to the Query is absent in the image, resulting in a caption like "Nothing." Including such cases would add valuable real-world scenarios to the dataset.

3.The impact of pre-training on this dataset for other Image Captioning tasks remains unknown. Further investigation is required to assess its potential benefits for related tasks.

**Reproducibility:**

4: Could mostly reproduce the results, but there may be some variation because of sample variance or minor variations in their interpretation of the protocol or method.

**Reviewer Confidence:**

3: Pretty sure, but there's a chance I missed something. Although I have a good feel for this area in general, I did not carefully check the paper's details, e.g., the math, experimental design, or novelty.

---

> ### Author Rebuttal · Authors · 2023-08-29
>
> > Limited number of models
>
> We prioritized the ablation studies over the validations of numerous models. We emphasize that  OFA and BLIP-2 were two state-of-the-art image captioning methods before submission. Furthermore, the effectiveness of query-based image captioning was verified using these two methods. The automatic evaluation metrics and the human evaluation in Table 3 and Table 4 proved that the introduction of queries helps captioning models to select the contexts from the images.
>
> > Lack mismatched cases
>
> Although the absence of the topic from images is interesting, discriminating the irrelevant topics  are well-known as notoriously difficult similar to the “no-answer” or “unanswerable” instances of the previous visual question answering dataset (e.g. Ray et al. [2]) and reading comprehension dataset of SQuAD v2. We hence consider it is an advanced ability for the current models to deal with mismatched cases and hence make them out-of-scope of the current dataset, concentrating on the annotations where topiced contexts surely exist in 360-degree images. We also believe we can develop the augmented pseudo mismatched cases with random topics and images with our annotations in the future work.
>
> > The impact of pre-training on this dataset for other tasks
>
> We designed QuIC-360° explicitly for fine-tuning models on the QuIC task. Using this dataset as a pre-training for other tasks is clearly outside the scope of this paper, so we do not think it is a reason to reject. We appreciate the reviewer's suggestion, but it would be helpful if you could tell us which task we should use it for.
>
> > Can this dataset(QuIC), like SA[0], use "points" or bounding boxes …
>
> In the problem setup of the query-based image captioning task, we take an image and a scene query as input and generate a caption corresponding to the query. So we cannot use any points or bounding boxes directly as a cue. Instead, it is possible to predict the bounding box from the scene query and image, and indirectly generate a caption based on the detected bounding box.
>
> Motivated by your insight, we implemented an additional query-based image captioning method using the state-of-the-art referring expression comprehension model of OFA and a Caption Anything [1] that internally uses SA [0]. First, the OFA (fine-tune with RefCOCO) predicts a bounding box from the scene query and image. Subsequently, the Caption Anything generates captions based on this predicted bounding box.
>
> We conducted additional experimental using this pipeline method and summarize the evaluation results as follows:
>
> [Setup]
> - Visual Grounding Model: OFA Large (fine-tuned with RefCOCO)
> - Caption Anything:
>   - Segmenter: sam_vit_h_4b8939.pth,
>   - Captioner: Salesforce/blip2-opt-2.7b
>
> [Metrics]
> - BLEU-4: 0.0246
> - METEOR: 0.0599
> - ROUGE: 0.2020
> - CIDEr: 0.0451
> - SPICE: 0.0347
> - CLIPScore: 0.6710
>
> We evaluated the SoTA model with  SA[0], but its performance was inferior to our methods. We will add this experiment in the appendices in the camera-ready version.
>
> We appreciate the constructive feedback and hope our clarifications address the concerns.
>
> [1] Caption Anything: Interactive Image Description with Diverse Multimodal Controls (Wang et al., 2023)
>
> [2] Question Relevance in VQA: Identifying Non-Visual And False-Premise Questions (Ray et al., 2016)

---

### Official Review · Reviewer_jdnx · 2023-08-05

**Soundness:** 3

**Excitement:**

3: Ambivalent: It has merits (e.g., it reports state-of-the-art results, the idea is nice), but there are key weaknesses (e.g., it describes incremental work), and it can significantly benefit from another round of revision. However, I won't object to accepting it if my co-reviewers champion it.

**Paper Topic And Main Contributions:**

In this paper the authors present a dataset/task that is focused on the generation of captions for images coming from 360° cameras. The technical novelty of the work is focused on the use of context words to drive to controlled caption generation process. The task is called Query-based Image Cap- tioning (QuIC) for 360-degree images. Usefully the work contributes a new novel dataset that comprises 3,940 360-degree images and 18,459 pairs of queries and captions annotated manually. The queryset included 34 items that were put together after the dataset was reviewed. Almost all queries were categories of items, however one query was the noun phrase what are they doing.

The authors applied their datasets to OFA and BLIP-2 both in terms of vanilla uses of these models, but also with variants that were fine tuned to the QuIC dataset in the normal way, and also variants that used the OFA and BLIP-2 architectures with the textual prompts.

The results do demonstrate that finetuning an architecture that is primed with the context word does outperform OFA and BLIP that have not received such fine tuning. Given the very different nature of the dataset, this is not surprising -- I would have been surprised if that was not the case. The evaluation is however high quality and will be useful for others to build upon.

**Questions For The Authors:**

Why do you feel that the model proposed and the approach taken controlled image captioning is motivated and well suited to the particular challenges of the 360° image says you are working with?
I do not claim to be an expert or have experience working with image captioning for 360° image data sets, but a question I would have is would it not be more relevant to generate captions that actually acknowledge and take advantage of the special properties of the 360° image. Specifically would it not be useful to generate captions that are aware of the special arrangement relative to the centre of the 360° image assuming it to be the front? Similar to this the extreme left and right of the image would assumed to be the back, and associated with this would be challenges around generating images relative to stitch together content. It strikes me that this would be a task that will be more natural for the 360° dataset, rather than just focusing on standard issues around lexical controllability.
Can you give more information on the Refer360 dataset since it's an important part of the test approach?
You mentioned that refer-test is a distinct statuses that was based on the Refer360 dataset, but then later you say that refer-test is built from a split of the main dataset. Can you provide clarification on exactly what this data subset is built from?




**Reasons To Accept:**

The proposed new task domain along with the associated data set is a valuable contribution for the controlled image caption and community. Also a systematic investigation of a particular architecture for controlled caption generation based on context specified as simple words and phrases also is part of a trend that has been seen in the community lately, but is none the less valuable and of benefit to the community.
The analysis itself on the data and the model is high-quality and worthy of publication.
The related work section is high quality and covers areas directly and indirectly related to the current publication.


**Reasons To Reject:**

My biggest problem with this paper is that it isn't clear whether the paper is really trying to propose a new task and associated data sets or a particular model for controlled image caption generation. I find that the model used based on individual context words for controlling the airport could be just as applicable to any other form of data and is in no way tied to the particular challenges associated with 360° images. Therefore the arguments made about the links between 360° datasets and this particular form of controllable image captioning do not hold up well. I would prefer to see a task that is more focused on the particular challenges with describing the 360° days assess rather than one that's broader in terms of specifying basic contacts that would be as applicable to any other image type.
In the Experiment section, the autbors state that {We validate that the model learning with the QuIC- 360◦ dataset surely enhances the controllability of scene captioning with baseline models.". This is surely self evident, i.e. that the use of a new day success with a particular property will provide better overall results when a testate assess with that same property is then used for the evaluation.

**Reproducibility:**

3: Could reproduce the results with some difficulty. The settings of parameters are underspecified or subjectively determined; the training/evaluation data are not widely available.

**Reviewer Confidence:**

3: Pretty sure, but there's a chance I missed something. Although I have a good feel for this area in general, I did not carefully check the paper's details, e.g., the math, experimental design, or novelty.

---

> ### Author Rebuttal · Authors · 2023-08-29
>
> > Focus on the particular challenges with 360° images
>
> Our task is scene captioning with 360-degree images and therefore we concentrate on the comprehension of scenes through images with textual scenery topics, not solving the 360-degree visionary issues such as distorted coordinates.  360-degree images used in this dataset contain richer contexts than the conventional images in existing image captioning datasets, and our study proposes a new image captioning task for scenes addressing these multiple contexts of 360° images. Furthermore, the experiment presented in Table 3 demonstrates that the query-based captioning method is more effective for tackling this challenge compared to the methods for the conventional image captioning task.
> We agree that it is also an interesting direction to leverage other properties of 360° images for future research. Our proposed 360° query-based image captioning dataset will offer a first step in this direction.
>
> > Clarification on Refer360 dataset and our refer-test split
>
> The refer-test split is a subset of QuIC-360° and contains images with a different source from the main dataset. We collected 3,800 images by crawling Flickr for the main dataset and  selected 140 images from the Refer-360 dataset for the refer-test split. For both dataset, we newly annotated captions via MTurk. While we mentioned in lines 466-468 that the images in the Refer-test split differ from the test split of the main dataset, this explanation might have been insufficient. We will add a clarification in our camera ready version.

---

### Meta-Review · Area_Chair_KkKz · 2023-09-18

**Recommendation:** 3

**Metareview:**

This paper presents a new task for image captioning which requires a model to generate captions based on (1) a 360-degree picture, (2) a controlling query. This task is novel and interesting to the community. However, reviewers have raised some concerns as well: (1) It is not clear the benefit of using a 360-degree picture instead of a standard picture, (2) The experiments section can be improved.

Specifically, reviewer 4vda has provided several good suggestions to improve the quality of the paper, including more baselines and more challenging examples.

---

### Decision · Program_Chairs · 2023-10-07

**Decision:**

Accept-Findings

**Comment:**

This paper presents a new task for image captioning which requires a model to generate captions based on (1) a 360-degree picture, (2) a controlling query. This task is novel and interesting to the community. However, reviewers have raised some concerns as well: (1) It is not clear the benefit of using a 360-degree picture instead of a standard picture, (2) The experiments section can be improved.

Specifically, reviewer 4vda has provided several good suggestions to improve the quality of the paper, including more baselines and more challenging examples.